

# Towards a standard model for research in agent-based modeling and simulation

Nuno Fachada[1], Vitor V. Lopes[2], Rui C. Martins[3] and Agostinho C. Rosa[1]

[1] Institute for Systems and Robotics, LARSyS, Instituto Superior Técnico, Universidade de Lisboa, Lisboa, Portugal
[2] Universidad de las Fuerzas Armadas-ESPE, Sangolquí, Ecuador
[3] Life and Health Sciences Research Institute, School of Health Sciences, University of Minho, Braga, Portugal

## ABSTRACT

Agent-based modeling (ABM) is a bottom-up modeling approach, where each entity of the system being modeled is uniquely represented as an independent decision-making agent. ABMs are very sensitive to implementation details. Thus, it is very easy to inadvertently introduce changes which modify model dynamics. Such problems usually arise due to the lack of transparency in model descriptions, which constrains how models are assessed, implemented and replicated. In this paper, we present PPHPC, a model which aims to serve as a standard in agent based modeling research, namely, but not limited to, conceptual model specification, statistical analysis of simulation output, model comparison and parallelization studies. This paper focuses on the first two aspects (conceptual model specification and statistical analysis of simulation output), also providing a canonical implementation of PPHPC. The paper serves as a complete reference to the presented model, and can be used as a tutorial for simulation practitioners who wish to improve the way they communicate their ABMs.

## INTRODUCTION

Agent-based modeling (ABM) is a bottom-up modeling approach, where each entity of the system being modeled is uniquely represented as an independent decision-making agent. When prompted to act, each agent analyzes its current situation (e.g., what resources are available, what other agents are in the neighborhood), and acts appropriately, based on a set of rules. These rules express knowledge or theories about the respective low-level components. The global behavior of the system is the result from the simple, self-organized local relationships between the agents (*Fachada, 2008*). As such, ABM is a useful tool in simulating and exploring systems that can be modeled in terms of interactions between individual entities, e.g., biological cell cultures, ants foraging for food or military units in a battlefield (*Macal & North, 2008*). In practice, ABM can be considered a variation of discrete-event simulation, since state changes occur at specific points in time (*Law, 2015*).

Corresponding author
Nuno Fachada,
nfachada@laseeb.org

Spatial agent-based models (SABMs) are a subset of ABMs in which a spatial topology defines how agents interact (*Shook, Wang & Tang, 2013*). For example, an agent may be limited to interact with agents located within a specific radius, or may only move to a near physical or geographical location (*Macal & North, 2010*). SABMs have been extensively used to study a range of phenomena in the biological and social sciences (*Isaac, 2011*; *Shook, Wang & Tang, 2013*).

ABMs are very sensitive to implementation details: the impact that seemingly unimportant aspects such as data structures, algorithms, discrete time representation, floating point arithmetic or order of events can have on results is tremendous (*Wilensky & Rand, 2007*; *Merlone, Sonnessa & Terna, 2008*). As such, it is very easy to inadvertently introduce changes which will alter model dynamics. These type of issues usually derive from a lack of transparency in model descriptions, which constrains how models are assessed and implemented (*Müller et al., 2014*). Conceptual models should be well specified and adequately described in order to be properly implemented and replicated (*Edmonds & Hales, 2003*; *Wilensky & Rand, 2007*).

The ODD protocol (Overview, Design concepts, Details) is currently one of the most widely used templates for making model descriptions more understandable and complete, providing a comprehensive checklist that covers virtually all the key features that can define a model (*Grimm et al., 2010*). It allows modelers to communicate their models using a natural language description within a prescriptive and hierarchical structure, aiding in model design and fostering in-depth model comprehension (*Müller et al., 2014*). It is the recommended approach for documenting models in the CoMSES Net Computational Model Library (*Rollins et al., 2014*). However, *Müller et al. (2014)* argue that no single model description standard can completely and throughly characterize a model by itself, suggesting that besides a structured natural language description such as ODD, the availability of a model's source code should be part of a minimum standard for model communication. Furthermore, the ODD protocol does not deal with models from a results or simulation output perspective, which means that an additional section for statistical analysis of results is often required. In practice, however, the situation is very different. While many ABMs have been published and simulation output analysis is a widely discussed subject matter (*Sargent, 1976*; *Kelton, 1997*; *Law, 2007*; *Nakayama, 2008*; *Law, 2015*), comprehensive inquiries concerning the output of ABM simulations are hard to find in the scientific literature.

In this paper, we present PPHPC (Predator-Prey for High-Performance Computing), a conceptual model which captures important characteristics of SABMs, such as agent movement and local agent interactions. It aims to serve as a standard in agent based modeling research, and was designed with several goals in mind:

1. Provide a basis for a tutorial on complete model specification and thorough simulation output analysis.

2. Investigate statistical comparison strategies for model replication (*Fachada et al., 2015a*).

3. Compare different implementations from a performance point of view, using different frameworks, programming languages, hardware and/or parallelization strategies, while maintaining statistical equivalence among implementations (*Fachada et al., 2015b*).

4. Test the influence of different pseudo-random number generators (PRNGs) on the statistical accuracy of simulation output.

This paper aims to fulfill the first of these goals, and is organized as follows. First, in 'Background,' we review several paradigmatic ABMs, as well as model description and analysis. Next, the 'Methodology' section is divided into five subsections, in which we: (a) formalize the conceptual model using the ODD protocol; (b) describe the canonical PPHPC realization implemented with the NetLogo ABM toolkit (*Wilensky, 1999*); (c) discuss how to select output focal measures; (d) explain how to collect and prepare data for statistical analysis; and, (e) propose how to analyze focal measures from a statistical point-of-view. In 'Results', statistical analysis of output of the NetLogo implementation is performed. A discussion on how these results can be utilized in additional investigations is undertaken in 'Discussion'. 'Conclusions' provides a global outline of what was accomplished in this paper.

## BACKGROUND

Several ABMs have been used for the purpose of modeling tutorials and/or model analysis and replication. Probably, the most well known standard ABM is the "StupidModel," which consists of a series of 16 pseudo-models of increasing complexity, ranging from simple moving agents to a full predator-prey-like model. It was developed by *Railsback, Lytinen & Grimm (2005)* as a teaching tool and template for real applications, as it includes a set of features commonly used in ABMs of real systems. It has been used to address a number of questions, including the comparison of ABM platforms (*Railsback, Lytinen & Jackson, 2006*; *Lytinen & Railsback, 2012*), model parallelization (*Lysenko & D'Souza, 2008*; *Tang & Wang, 2009*), analysis of toolkit feasibility (*Standish, 2008*) and/or creating models as compositions of micro-behaviors (*Kahn, 2007*). The "StupidModel" series has been criticized for having some atypical elements and ambiguities (*Lytinen & Railsback, 2012*), reasons which lead *Isaac (2011)* to propose a reformulation to address these and other issues. However, its multiple versions and user-interface/visualization goals limit the series appeal as a pure computational model for the goals described in the introduction.

Other paradigmatic models which have been recurrently used, studied and replicated include Sugarscape (*Epstein & Axtell, 1996*; *Axtell et al., 1996*; *Bigbee, Cioffi-Revilla & Luke, 2007*; *D'Souza, Lysenko & Rahmani, 2007*; *Lysenko & D'Souza, 2008*), Heatbugs (*Wilensky, 2004*; *Sallach & Mellarkod, 2005*; *Goldsby & Pancerella, 2013*), Boids (*Reynolds, 1987*; *Reynolds, 2006*; *Goldsby & Pancerella, 2013*) and several interpretations of proto-typical predator-prey models (*Smith, 1991*; *Hiebeler, 1994*; *Wilensky, 1997*; *Tatara et al., 2006*; *Ottino-Loffler, Rand & Wilensky, 2007*; *Ginovart, 2014*). Nonetheless, there is a lack of formalization and in-depth statistical analysis of simulation output in most of these implementations, often leading to model assessment and replication difficulties (*Edmonds*

*& Hales, 2003*; *Wilensky & Rand, 2007*). This might not come as a surprise, as most models are not implemented with replication in mind.

Many models are not adequately analyzed with respect to their output data, often due to improper design of simulation experiments. Consequently, authors of such models can be at risk of making incorrect inferences about the system being studied (*Law, 2007*). A number of papers and books have been published concerning the challenges, pitfalls and opportunities of using simulation models and adequately analyzing simulation output data. In one of the earliest articles on the subject, *Sargent (1976)* demonstrates how to obtain point estimates and confidence intervals for steady state means of simulation output data using a number of different methodologies. Later, *Law (1983)* presented a state-of-the-art survey on statistical analyses for simulation output data, addressing issues such as start-up bias and determination of estimator accuracy. This survey was updated several times over the years, e.g., *Law (2007)*, where Law discusses the duration of transient periods before steady state settles, as well as the number of replications required for achieving a specific level of estimator confidence. In *Kelton (1997)*, the author describes methods to help design the runs for simulation models and interpreting their output using statistical methods, also dealing with related problems such as model comparison, variance reduction or sensitivity estimation. A comprehensive exposition of these and other important topics of simulation research is presented in the several editions of "Simulation Modeling and Analysis" by Law and Kelton, and its latest edition (*Law, 2015*) is used as a starting point for the analysis described in 'Methodology' and conducted in 'Results.'

# METHODOLOGY

## Overview, design concepts and details of PPHPC

Here we describe the PPHPC model using the ODD protocol (*Grimm et al., 2010*). Time-dependent state variables are represented with uppercase letters, while constant state variables and parameters are denoted by lowercase letters. The $U(a, b)$ expression equates to a random integer within the closed interval $[a, b]$ taken from the uniform distribution.

### *Purpose*

The purpose of PPHPC is to serve as a standard model for studying and evaluating SABM implementation strategies. It is a realization of a predator-prey dynamic system, and captures important characteristics of SABMs, such as agent movement and local agent interactions. The model can be implemented using substantially different approaches that ensure statistically equivalent qualitative results. Implementations may differ in aspects such as the selected system architecture, choice of programming language and/or agent-based modeling framework, parallelization strategy, random number generator, and so forth. By comparing distinct PPHPC implementations, valuable insights can be obtained on the computational and algorithmical design of SABMs in general.

**Table 1  Model state variables by entity.** Where applicable, the *s* and *w* designations correspond to prey (*sheep*) and predator (*wolf*) agent types, respectively.

| Entity | State variable | Symbol | Range |
|---|---|---|---|
| Agents | Type | $t$ | $s, w$ |
| | Energy | $E$ | $1, 2, \ldots$ |
| | Horizontal position in grid | $X$ | $0, 1, \ldots, x_{env} - 1$ |
| | Vertical position in grid | $Y$ | $0, 1, \ldots, y_{env} - 1$ |
| | Energy gain from food | $g^s, g^w$ | $0, 1, \ldots$ |
| | Energy loss per turn | $l^s, l^w$ | $0, 1, \ldots$ |
| | Reproduction threshold | $r_T^s, r_T^w$ | $1, 2, \ldots$ |
| | Reproduction probability | $r_P^s, r_P^w$ | $0, 1, \ldots, 100$ |
| Grid cells | Horizontal position in grid | $x$ | $0, 1, \ldots, x_{env} - 1$ |
| | Vertical position in grid | $y$ | $0, 1, \ldots, y_{env} - 1$ |
| | Countdown | $C$ | $0, 1, \ldots, c_r$ |
| Environment | Horizontal size | $x_{env}$ | $1, 2, \ldots$ |
| | Vertical size | $y_{env}$ | $1, 2, \ldots$ |
| | Restart | $c_r$ | $1, 2, \ldots$ |

### Entities, state variables, scales

The PPHPC model is composed of three entity classes: *agents*, *grid cells* and *environment*. Each of these entity classes is defined by a set of state variables, as shown in Table 1. All state variables explicitly assume integer values to avoid issues with the handling of floating-point arithmetic on different programming languages and/or processor architectures.

The *t* state variable defines the *agent* type, either *s* (*sheep*, i.e. prey) or *w* (*wolf*, i.e. predator). The only behavioral difference between the two types is in the feeding pattern: while prey consume passive cell-bound food, predators consume prey. Other than that, prey and predators may have different values for other state variables, as denoted by the superscripts *s* and *w*. Agents have an energy state variable, *E*, which increases by $g^s$ or $g^w$ when feeding, decreases by $l^s$ or $l^w$ when moving, and decreases by half when reproducing. When energy reaches zero, the agent is removed from the simulation. Agents with energy higher than $r_T^s$ or $r_T^w$ may reproduce with probability given by $r_P^s$ or $r_P^w$. The grid position state variables, *X* and *Y*, indicate which cell the agent is located in. There is no conceptual limit on the number of agents that can exist during the course of a simulation run.

Instances of the *grid cell* entity class can be thought of the place or neighborhood where agents act, namely where they try to feed and reproduce. Agents can only interact with other agents and resources located in the same grid cell. Grid cells have a fixed grid position, $(x, y)$, and contain only one resource, cell-bound food (*grass*), which can be consumed by prey, and is represented by the countdown state variable *C*. The *C* state variable specifies the number of iterations left for the cell-bound food to become available. Food becomes available when $C = 0$, and when a prey consumes it, *C* is set to $c_r$.

The set of all grid cells forms the *environment* entity, a toroidal square grid where the simulation takes place. The environment is defined by its size, $(x_{env}, y_{env})$, and by the restart parameter, $c_r$.

Spatial extent is represented by the aforementioned square grid, of size $(x_{env}, y_{env})$, where $x_{env}$ and $y_{env}$ are positive integers. Temporal extent is represented by a positive integer $m$, which represents the number of discrete simulation steps or iterations. Spatial and temporal scales are merely virtual, i.e. they do not represent any real measure.

### Process overview and scheduling

Algorithm 1 describes the simulation schedule and its associated processes. Execution starts with an initialization process, `Init()`, where a predetermined number of agents are randomly placed in the simulation environment. Cell-bound food is also initialized at this stage.

After initialization, and to get the simulation state at iteration zero, outputs are gathered by the `GetStats()` process. The scheduler then enters the main simulation loop, where each iteration is sub-divided into four steps: (1) agent movement ; (2) food growth in grid cells; (3) agent actions ; and, (4) gathering of simulation outputs. State variables

---

**Algorithm 1** Main simulation algorithm. **for** loops can be processed in *any order* or in *random order*. In terms of expected dynamic behavior, the former means the order is not relevant, while the latter specifies loop iterations should be explicitly shuffled.

---

1: INIT()
2: GETSTATS()
3: $i \leftarrow 1$
4: **for** $i <= m$ **do**
5:     **for each** agent **do**           ▷ Any order
6:         MOVE()
7:     **end for**
8:     **for each** grid cell **do**         ▷ Any order
9:         GROWFOOD()
10:     **end for**
11:     **for each** agent **do**          ▷ Random order
12:         ACT()
13:     **end for**
14:     GETSTATS()
15:     $i \leftarrow i + 1$
16: **end for**

---

are asynchronously updated, i.e. they are assigned a new value as soon as this value is calculated by a process (e.g., when an agent gains energy by feeding).

### Design concepts

*Basic principles.* The general concepts of this model are based on well studied predator-prey dynamics, initially through analytical approaches (*Lotka, 1925*; *Volterra, 1926*), and later using agent-based models (*Smith, 1991*). However, PPHPC is designed so that it can be correctly implemented using diverse computational approaches. Realizations of this

model can provide valuable information on how to better implement SABMs on different computing architectures, namely parallel ones. In particular, they may shown the impact of different parallelization strategies on simulation performance.

*Emergence.* The model is characterized by oscillations in the population of both prey and predator, as well as in the available quantity of cell-bound food. Typically, a peak of predator population occurs slightly after a peak in prey population size, while quantity of cell-bound food is approximately in "phase opposition" with the prey's population size.

*Sensing.* Agents can sense the presence of food in the grid cell in which they are currently located. This means different thing for prey and predators. Prey agents can read the local grid cell $C$ state variable, which if zero, means there is food available. Predator agents can determine the presence of prey agents.

*Interaction.* Agents interact with sources of food present in the grid cell they are located in.

*Stochasticity.* The following processes are random: (a) initialization of specific state variables; (b) agent movement; (c) the order in which agents act; and, (d) agent reproduction.

*Observation.* The following vector is collected in the `GetStats()` process, where $i$ refers to the current iteration:

$$\mathbf{O}_i = (P_i^s, P_i^w, P_i^c, \overline{E}_i^s, \overline{E}_i^w, \overline{C}_i)$$

$P_i^s$ and $P_i^w$ refer to the total prey and predator population counts, respectively, while $P_i^c$ holds the quantity of available cell-bound food. $\overline{E}_i^s$ and $\overline{E}_i^w$ contain the mean energy of prey and predator populations. Finally, $\overline{C}_i$ refers to the mean value of the $C$ state variable in all grid cells.

### Initialization

The initialization process begins by instantiating the *environment* entity, a toroidal square grid, and filling it with $x_{env} \times y_{env}$ grid cells. The initial value of the countdown state variable in each grid cell, $C_0$, is set according to Eq. (1),

$$C_0 = \begin{cases} U(1, c_r), & \text{if } c_0 = 0 \\ 0, & \text{if } c_0 = 1, \end{cases} \quad \text{with } c_0 = U(0, 1). \tag{1}$$

In other words, cell-bound food is initially available with 50% probability. If not available, the countdown state variable is set to a random value between 1 and $c_r$.

The initial value of the state variables for each agent is determined according to Eqs. (2) and (3).

$$E_0 = U(1, 2g), \quad \text{with } g \in \{g^s, g^w\} \tag{2}$$

$$(X_0, Y_0) = \big(U(0, x_{env} - 1), U(0, y_{env} - 1)\big). \tag{3}$$

### Submodels

As stated in *Process overview and scheduling*, each iteration of the main simulation loop is sub-divided into four steps, described in the following paragraphs.

*Move().* In step 1, agents `Move()`, in any order, within a Von Neumann neighborhood, i.e. up, down, left, right or stay in the same cell, with equal probability. Agents lose $l^s$ or $l^w$ units of energy when they move, even if they stay in the same cell; if energy reaches zero, the agent dies and is removed from the simulation.

*GrowFood().* In step 2, during the `GrowFood()` process, each grid cell checks if $C = 0$ (meaning there is food available). If $C > 0$ it is decremented by one unit. Equation (4) summarizes this process.

$$C_i = \max(C_{i-1} - 1, 0). \tag{4}$$

*Act().* In step 3, agents `Act()` in explicitly random order, i.e. the agent list should be shuffled before the agents have a chance to act. The `Act()` process is composed of two sub-actions: `TryEat()` and `TryReproduce()`. The `Act()` process is atomic, i.e. once called, both `TryEat()` and `TryReproduce()` must be performed; this implies that prey agents may be killed by predators before or after they have a chance of calling `Act()`, but not during the call.

*TryEat().* Agents can only interact with sources of food present in the grid cell they are located in. Predator agents can kill and consume prey agents, removing them from the simulation. Prey agents can consume cell-bound food, resetting the local grid cell $C$ state variable to $c_r$. A predator can consume one prey per iteration, and a prey can only be consumed by one predator. Agents who act first claim the food resources available in the local grid cell. Feeding is automatic: if the resource is there and no other agent has yet claimed it, the agent will consume it. Moreover, only one prey can consume the local cell-bound food if available (i.e. if $C = 0$). When an agent successfully feeds, its energy $E$ is incremented by $g^s$ or $g^w$, depending on whether the agent is a prey or a predator, respectively.

*TryReproduce().* If the agent's energy, $E$, is above its species reproduction threshold, $r_T^s$ or $r_T^w$, then reproduction will occur with probability given by the species reproduction probability, $r_P^s$ or $r_P^w$, as shown in Algorithm 2. When an agent successfully reproduces, its energy is divided (using integer division) with its offspring. The offspring is placed in the same grid cell as his parent, but can only take part in the simulation in the next iteration. More specifically, newly born agents cannot `Act()`, nor be acted upon. The latter implies that newly born prey cannot be consumed by predators in the current iteration. Agents immediately update their energy if they successfully feed and/or reproduce.

*Parameterization.* Model parameters can be qualitatively separated into size-related and dynamics-related parameters, as shown in Table 2. Although size-related parameters also influence model dynamics, this separation is useful for parameterizing simulations.

**Algorithm 2** Agent reproduction.

```
function TRYREPRODUCE()
    if E > r_T then
        if U(0, 99) < r_P then
            E^child ← E/2                    ▷ Integer division
            E ← E − E^child
            NEWAGENT(t, E^child, X, Y)
        end if
    end if
end function
```

**Table 2** Size-related and dynamics-related model parameters.

| Type | Parameter | Symbol |
|------|-----------|--------|
| Size | Environment size | $x_{env}, y_{env}$ |
| | Initial agent count | $P_0^s, P_0^w$ |
| | Number of iterations | $m$ |
| Dynamics | Energy gain from food | $g^s, g^w$ |
| | Energy loss per turn | $l^s, l^w$ |
| | Reproduction threshold | $r_T^s, r_T^w$ |
| | Reproduction probability | $r_P^s, r_P^w$ |
| | Cell food restart | $c_r$ |

**Table 3** A selection of initial model sizes.

| Size | $x_{env} \times y_{env}$ | $P_0^s$ | $P_0^w$ |
|------|--------------------------|---------|---------|
| 100 | $100 \times 100$ | 400 | 200 |
| 200 | $200 \times 200$ | 1,600 | 800 |
| 400 | $400 \times 400$ | 6,400 | 3,200 |
| 800 | $800 \times 800$ | 25,600 | 12,800 |
| 1,600 | $1,600 \times 1,600$ | 102,400 | 51,200 |
| ⋮ | ⋮ | ⋮ | ⋮ |

Concerning size-related parameters, more specifically, the grid size, we propose a base value of $100 \times 100$, associated with 400 prey and 200 predators. Different grid sizes should have proportionally assigned agent population sizes, as shown in Table 3. In other words, there are no changes in the agent density nor the ratio between prey and predators.

For the dynamics-related parameters, we propose two sets of parameters, Table 4, which generate two distinct dynamics. The second parameter set typically yields more than twice the number of agents than the first parameter set. Matching results with runs based on distinct parameters is necessary in order to have a high degree of confidence in the similarity of different implementations (*Edmonds & Hales, 2003*). While many more combinations

**Table 4** Dynamics-related parameter sets.

| Parameter | Symbol | Set 1 | Set 2 |
|---|---|---|---|
| Prey energy gain from food | $g^s$ | 4 | 30 |
| Prey energy loss p/turn | $l^s$ | 1 | 1 |
| Prey reprod. threshold | $r_T^s$ | 2 | 2 |
| Prey reprod. probability | $r_P^s$ | 4 | 10 |
| Predator energy gain from food | $g^w$ | 20 | 10 |
| Predator energy loss p/turn | $l^w$ | 1 | 1 |
| Predator reprod. threshold | $r_T^w$ | 2 | 2 |
| Predator reprod. probability | $r_P^w$ | 5 | 5 |
| Cell food restart | $c_r$ | 10 | 15 |

of parameters can be experimented with this model, these two sets are the basis for testing and comparing PPHPC implementations. We will refer to a combination of model size and parameter set as "size@set," e.g., 400@1 for model size 400, parameter set 1.

While simulations of the PPHPC model are essentially non-terminating,[1] the number of iterations, $m$, is set to 4,000, as it allows to analyze steady-state behavior for all the parameter combinations discussed here.

[1] A non-terminating simulation is one for which there is no natural event to specify the length of a run (*Law, 2015*).

## A NetLogo implementation

NetLogo is a well-documented programming language and modeling environment for ABMs, focused on both research and education. It is written in Scala and Java and runs on the Java Virtual Machine (JVM). It uses a hybrid interpreter and compiler that partially compiles ABM code to JVM bytecode (*Sondahl, Tisue & Wilensky, 2006*). It comes with powerful built-in procedures and is relatively easy to learn, making ABMs more accessible to researchers without programming experience (*Martin et al., 2012*). Advantages of having a NetLogo version include real-time visualization of simulation, pseudo-code like model descriptions, simplicity in changing and testing different model aspects and parameters, and command-line access for batch runs and cycling through different parameter sets, even allowing for multithreaded simultaneous execution of multiple runs. A NetLogo reference implementation is also particularly important as a point of comparison with other ABM platforms (*Isaac, 2011*).

The NetLogo implementation of PPHPC, Fig. 1, is based on NetLogo's own *Wolf Sheep Predation* model (*Wilensky, 1997*), considerably modified to follow the ODD discussed in the previous section. Most NetLogo models will have at least a *setup* procedure, to set up the initial state of the simulation, and a *go* procedure to make the model run continuously (*Wilensky, 2014*). The `Init()` and `GetStats()` processes (lines 1 and 2 of algorithm 1) are defined in the *setup* procedure, while the main simulation loop is implemented in the *go* procedure. The latter has an almost one-to-one relation with its pseudo-code counterpart in Algorithm 1. By default, NetLogo shuffles agents before issuing them orders, which fits naturally into the model ODD. The implementation is available at https://github.com/fakenmc/pphpc/tree/netlogo.

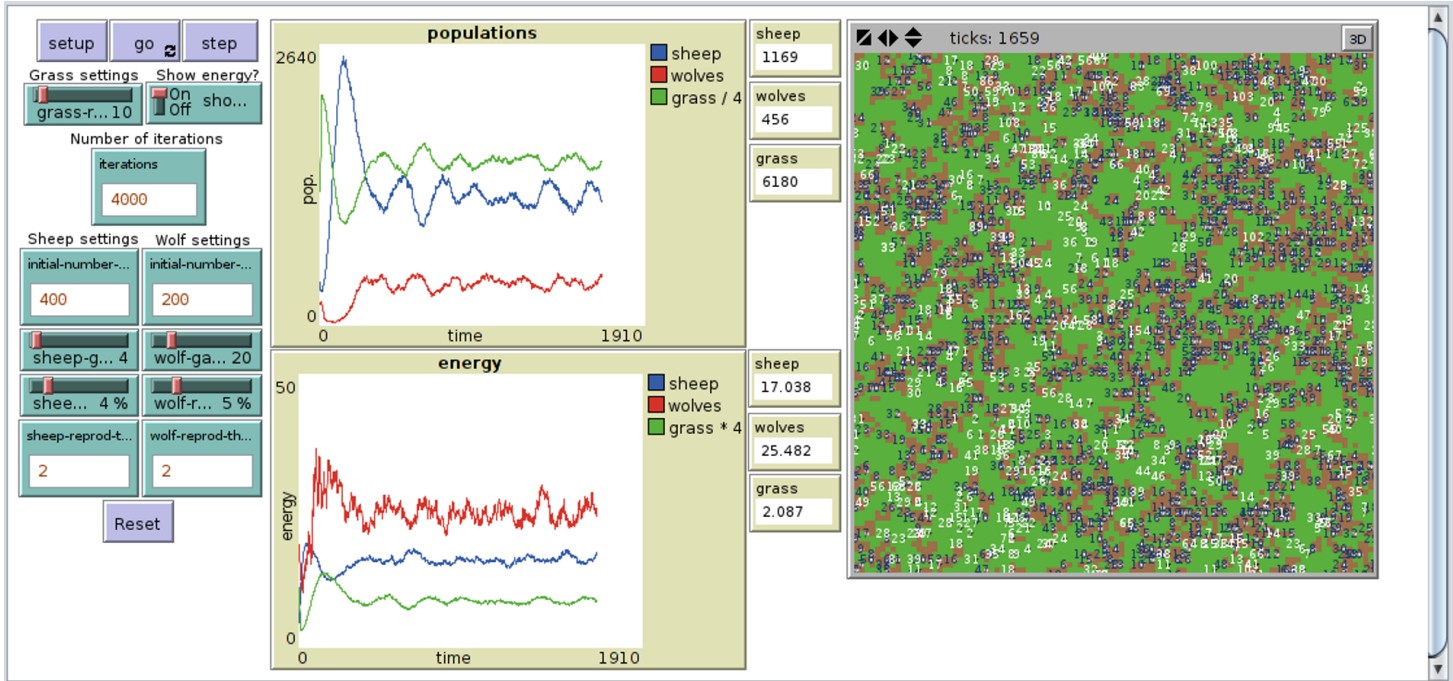

**Figure 1  NetLogo implementation of the PPHPC model.**

## Selection of focal measures

In order to analyze the output of a simulation model from a statistical point-of-view, we should first select a set of focal measures (FMs) which summarize each output. *Wilensky & Rand (2007)* use this approach in the context of statistical comparison of replicated models. Typically, FMs consist of long-term or steady-state means. However, being limited to analyze average system behavior can lead to incorrect conclusions (*Law, 2015*). Consequently, other measures such as proportions or extreme values can be used to assess model behavior. In any case, the selection of FMs is an empirical exercise and is always dependent of the model under study. A few initial runs are usually required in order to perform this selection.

For the PPHPC model, the typical output of a simulation run is shown in Fig. 2 for size 400 and both parameter sets. In both cases, all outputs undergo a transient stage and tend to stabilize after a certain number of iterations, entering steady-state. For other sizes, the situation is similar apart from a vertical scaling factor. Outputs display pronounced extreme values in the transient stage, while circling around a long-term mean and approximately constant standard deviation in the steady-state phase. This standard deviation is an important feature of the outputs, as it marks the overall variability of the predator-prey cycles. Having this under consideration, six statistics, described in Table 5, were selected for each output. Considering there are six outputs, a total of 36 FMs are analyzed for the PPHPC model.

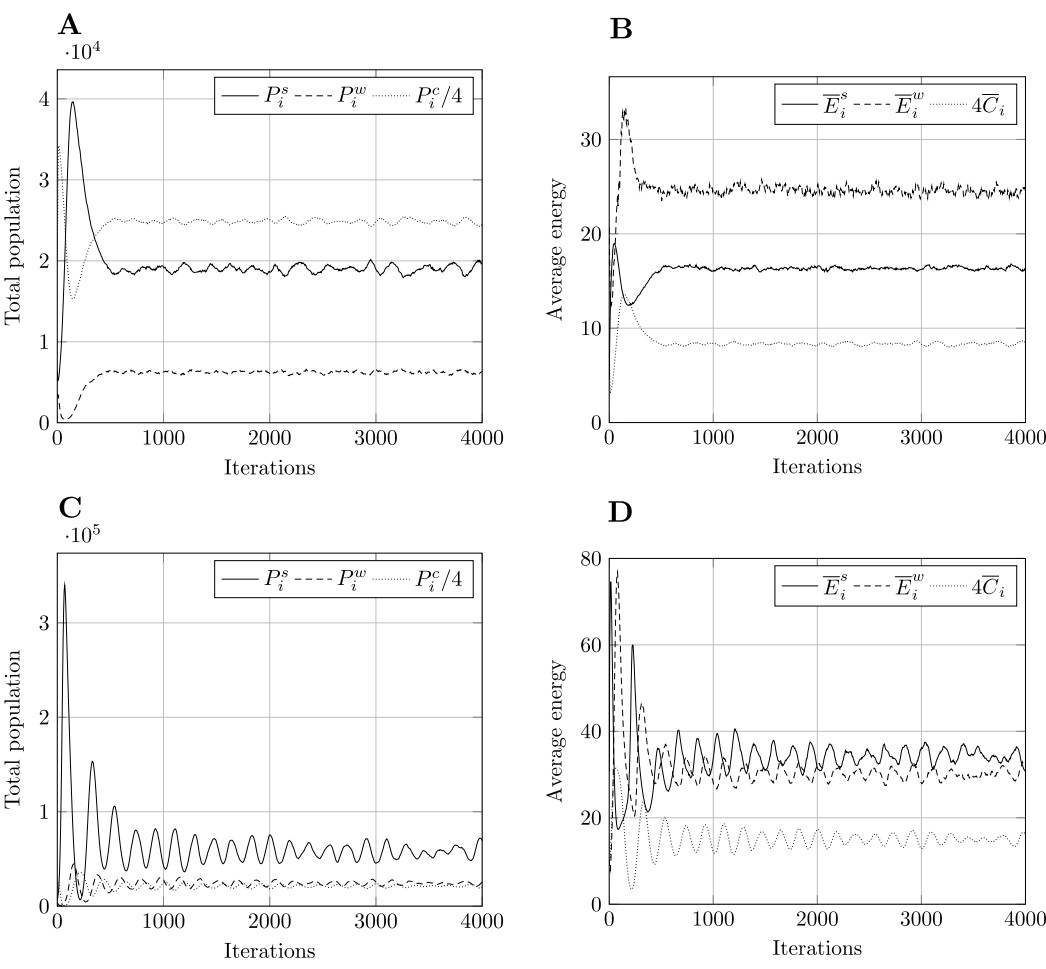

**Figure 2 Typical model output for model size 400.** Other model sizes have outputs which are similar, apart from a vertical scaling factor. $P_i$ refers to total population, $\overline{E}_i$ to mean energy and $\overline{C}_i$ to mean value of the countdown state variable, $C$. Superscript $s$ relates to prey, $w$ to predators, and $c$ to cell-bound food. $P_i^c$ and $\overline{C}_i$ are scaled for presentation purposes. (A) Population, param. set 1. (B) Energy, param. set 1. (C) Population, param. set 2. (D) Energy, param. set 2.

**Table 5 Statistical summaries for each output $X$.** $X_i$ is the value of $X$ at iteration $i$, $m$ denotes the last iteration, and $l$ corresponds to the iteration separating the transient and steady-state stages.

| Statistic | Description |
|---|---|
| $\max_{0 \le i \le m} X_i$ | Maximum value. |
| $\arg\max_{0 \le i \le m} X_i$ | Iteration where maximum value occurs. |
| $\min_{0 \le i \le m} X_i$ | Minimum value. |
| $\arg\min_{0 \le i \le m} X_i$ | Iteration where minimum value occurs. |
| $\overline{X}^{ss} = \sum_{i=l+1}^{m} X_i / (m-l)$ | Steady-state mean. |
| $S^{ss} = \sqrt{\dfrac{\sum_{i=l+1}^{m}(X_i - \overline{X}_{ss})^2}{m-l-1}}$ | Steady-state sample standard deviation. |

**Table 6 Values of a generic simulation output (under 'Iterations') for $n$ replications of $m$ iterations each (plus iteration 0, i.e. the initial state), and the respective FMs (under 'Focal measures').** Values along columns are IID.

| Rep. | Iterations | | | | | Focal measures | | | | | |
|---|---|---|---|---|---|---|---|---|---|---|---|
| 1 | $X_{10}$ | $X_{11}$ | ... | $X_{1,m-1}$ | $X_{1,m}$ | $\max X_1$ | $\arg\max X_1$ | $\min X_1$ | $\arg\min X_1$ | $\overline{X}_1^{ss}$ | $S_1^{ss}$ |
| 2 | $X_{20}$ | $X_{21}$ | ... | $X_{2,m-1}$ | $X_{2,m}$ | $\max X_2$ | $\arg\max X_2$ | $\min X_2$ | $\arg\min X_2$ | $\overline{X}_2^{ss}$ | $S_2^{ss}$ |
| $\vdots$ | $\vdots$ | $\vdots$ | $\vdots$ | $\vdots$ | $\vdots$ | $\vdots$ | $\vdots$ | $\vdots$ | $\vdots$ | $\vdots$ | $\vdots$ |
| $n$ | $X_{n0}$ | $X_{n1}$ | ... | $X_{n,m-1}$ | $X_{n,m}$ | $\max X_n$ | $\arg\max X_n$ | $\min X_n$ | $\arg\min X_n$ | $\overline{X}_n^{ss}$ | $S_n^{ss}$ |

## Collecting and preparing data for statistical analysis

Let $X_{j0}, X_{j1}, X_{j2}, \ldots, X_{jm}$ be an output from the $j^{th}$ simulation run (rows under 'Iterations' in Table 6). The $X_{ji}$'s are random variables that will, in general, be neither independent nor identically distributed (*Law, 2015*), and as such, are not adequate to be used directly in many formulas from classical statistics (which are discussed in the next section). On the other hand, let $X_{1i}, X_{2i}, \ldots, X_{ni}$ be the observations of an output at iteration $i$ for $n$ runs (columns under 'Iterations' in Table 6), where each run begins with the same initial conditions but uses a different stream of random numbers as a source of stochasticity. The $X_{ji}$'s will now be independent and identically distributed (IID) random variables, to which classical statistical analysis can be applied. However, individual values of the output $X$ at some iteration $i$ are not representative of $X$ as a whole. Thus, we use the selected FMs as representative summaries of an output, as shown in Table 6, under 'Focal measures.' Taken column-wise, the observations of the FMs are IID (because they are obtained from IID replications), constituting a *sample* prone to statistical analysis.

Regarding steady-state measures, $\overline{X}^{ss}$ and $S^{ss}$, care must be taken with initialization bias, which may cause substantial overestimation or underestimation of the long-term performance (*Sanchez, 1999*). Such problems can be avoided by discarding data obtained during the initial transient period, before the system reaches steady-state conditions. The simplest way of achieving this is to use a fixed truncation point, $l$, for all runs with the same initial conditions, selected such that: (a) it systematically occurs after the transient state; and, (b) it is associated with a round and clear value, which is easier to communicate (*Sanchez, 1999*). *Law (2015)* suggests the use of Welch's procedure (*Welch, 1981*) in order to empirically determine $l$. Let $\overline{X}_0, \overline{X}_1, \overline{X}_2, \ldots, \overline{X}_m$ be the averaged process taken column-wise from Table 6 (columns under 'Iterations'), such that $\overline{X}_i = \sum_{j=1}^{n} X_{ji}/n$ for $i = 0, 1, \ldots, m$. The averaged process has the same transient mean curve as the original process, but its variance is reduced by a factor of $n$. A low-pass filter can be used to remove short-term fluctuations, leaving the long-term trend of interest, allowing us to visually determine a value of $l$ for which the averaged process seems to have converged. A moving average approach can be used for filtering:

$$\overline{X}_i(w) = \begin{cases} \dfrac{\sum_{s=-w}^{w} \overline{X}_{i+s}}{2w+1} & \text{if } i = w+1, \ldots, m-w \\ \dfrac{\sum_{s=-(i-1)}^{i-1} \overline{X}_{i+s}}{2i-1} & \text{if } i = 1, \ldots, w \end{cases} \tag{5}$$

where $w$, the *window*, is a positive integer such that $w \leqslant \lfloor m/4 \rfloor$. This value should be large enough such that the plot of $\overline{X}_i(w)$ is moderately smooth, but not any larger. A more in-depth discussion of this procedure is available in *Welch (1981)* and *Law (2015)*.

## Statistical analysis of focal measures

Let $Y_1, Y_2, \ldots, Y_n$ be IID observations of some FM with finite population mean $\mu$ and finite population variance $\sigma^2$ (i.e. any column under 'Focal measures' in Table 6). Then, as described by *Law (2007)* and *Law (2015)*, unbiased point estimators for $\mu$ and $\sigma^2$ are given by

$$\overline{Y}(n) = \frac{\sum_{j=1}^{n} Y_j}{n} \tag{6}$$

and

$$S^2(n) = \frac{\sum_{j=1}^{n} [Y_j - \overline{Y}(n)]^2}{n-1} \tag{7}$$

respectively.

Another common statistic usually determined for a given FM is the confidence interval (CI) for $\overline{Y}(n)$, which can be defined in several different ways. The $t$-distribution CI is commonly used for this purpose (*Law, 2007*; *Law, 2015*), although it has best coverage for normally distributed samples, which is often not the case for simulation models in general (*Sargent, 1976*; *Law, 2015*) and agent-based models in particular (*Helbing & Balietti, 2012*). If samples are drawn from populations with multimodal, discrete or strongly skewed distributions, the usefulness of $t$-distribution CIs is further reduced. While there is not much to do in the case of multimodal distributions, *Law (2015)* proposes the use of the CI developed by *Willink (2005)*, which takes distribution skewness into account. Furthermore, CIs for discrete distributions are less studied and usually assume data follows a binomial distribution, presenting some issues of its own (*Brown, Cai & DasGupta, 2001*). As suggested by *Radax & Rengs (2010)*, we focus on providing a detailed assessment of the distributional properties of the different FMs, namely whether they are sufficiently "normal" such that normality-assuming (parametric) statistical techniques can be applied, not only for CI estimation, but also for model comparison purposes.

The normality of a data set can be assessed graphically or numerically (*Park, 2008*). The former approach is intuitive, lending itself to empirical interpretation by providing a way to visualize how random variables are distributed. The latter approach is a more objective and quantitative form of assessing normality, providing summary statistics and/or statistics tests of normality. In both approaches, specific methods can be either descriptive or theory-driven, as shown in Table 7.

For this study we chose one method of each type, as shown in boldface in Table 7. This approach not only provides a broad overview of the distribution under study, but is also important because no single method can provide a complete picture of the distribution.

Under the graphical methods umbrella, a **histogram** shows the approximate distribution of a data set, and is built by dividing the range of values into a sequence of

**Table 7 Methods for assessing the normality of a data set, adapted from *Park (2008)*.** Boldface methods are used in this study.

|  | Graphical methods | Numerical methods |
|---|---|---|
| Descriptive | **Histogram**, Box plot, Dot plot | **Skewness**, Kurtosis |
| Theory-driven | **Q–Q plot**, P-P plot | **Shapiro-Wilk**, Anderson-Darling, Cramer-von Mises, Kolmogorov-Smirnov, Jarque-Bera and other tests |

intervals (*bins*), and counting how many values fall in each interval. A **Q–Q plot** compares the distribution of a data set with a specific theoretical distribution (e.g., the normal distribution) by plotting their quantiles against each other (thus "Q–Q"). If the two distributions match, the points on the plot will approximately lie on the $y = x$ line. While a histogram gives an approximate idea of the overall distribution, the Q–Q plot is more adequate to see how well a theoretical distribution fits the data set.

Concerning numerical methods, **Skewness** measures the degree of symmetry of a probability distribution about its mean, and is a commonly used metric in the analysis of simulation output data (*Sargent, 1976*; *Nakayama, 2008*; *Law, 2015*). If skewness is positive, the distribution is skewed to the right, and if negative, the distribution is skewed to the left. Symmetric distributions have zero skewness, however, the converse is not necessarily true, e.g., skewness will also be zero if both tails of an asymmetric distribution account for half the total area underneath the probability density function. In the case of theory-driven numerical approaches, we select the **Shapiro-Wilk** (SW) test (*Shapiro & Wilk, 1965*), as it has been shown to be more effective when compared to several other normality tests (*Razali & Wah, 2011*). We focus on the *p*-value of this test (instead of the test's own *W* statistic), as it is an easily interpretable measure. The null-hypothesis of this test is that the data set, or sample, was obtained from a normally distributed population. If the *p*-value is greater than a predetermined significance level $\alpha$, usually 0.01 or 0.05, then the null hypothesis cannot be rejected. Conversely, a *p*-value less than $\alpha$ implies the rejection of the null hypothesis, i.e., that the sample was not obtained from a normally distributed population.

# RESULTS

A total of 30 replications, $r = 1, \ldots, 30$, were performed with NetLogo 5.1.0 for each combination of model sizes (Table 3) and parameters sets (Table 4). Each replication $r$ was performed with a PRNG seed obtained by taking the MD5 checksum of $r$ and converting the resulting hexadecimal string to a 32-bit integer (the maximum precision accepted by NetLogo), guaranteeing some independence between seeds, and consequently, between replications. The list of seeds is provided in Table S1.

## Determining the steady-state truncation point

Using Welch's method, we smoothed the averaged outputs using a moving average filter with $w = 10$. Having experimented with other values, $w = 10$ seemed to be a good compromise between rough and overly smooth plots. Fig. 3 shows results for model size 400 and both parameter sets. Following the recommendations described in section

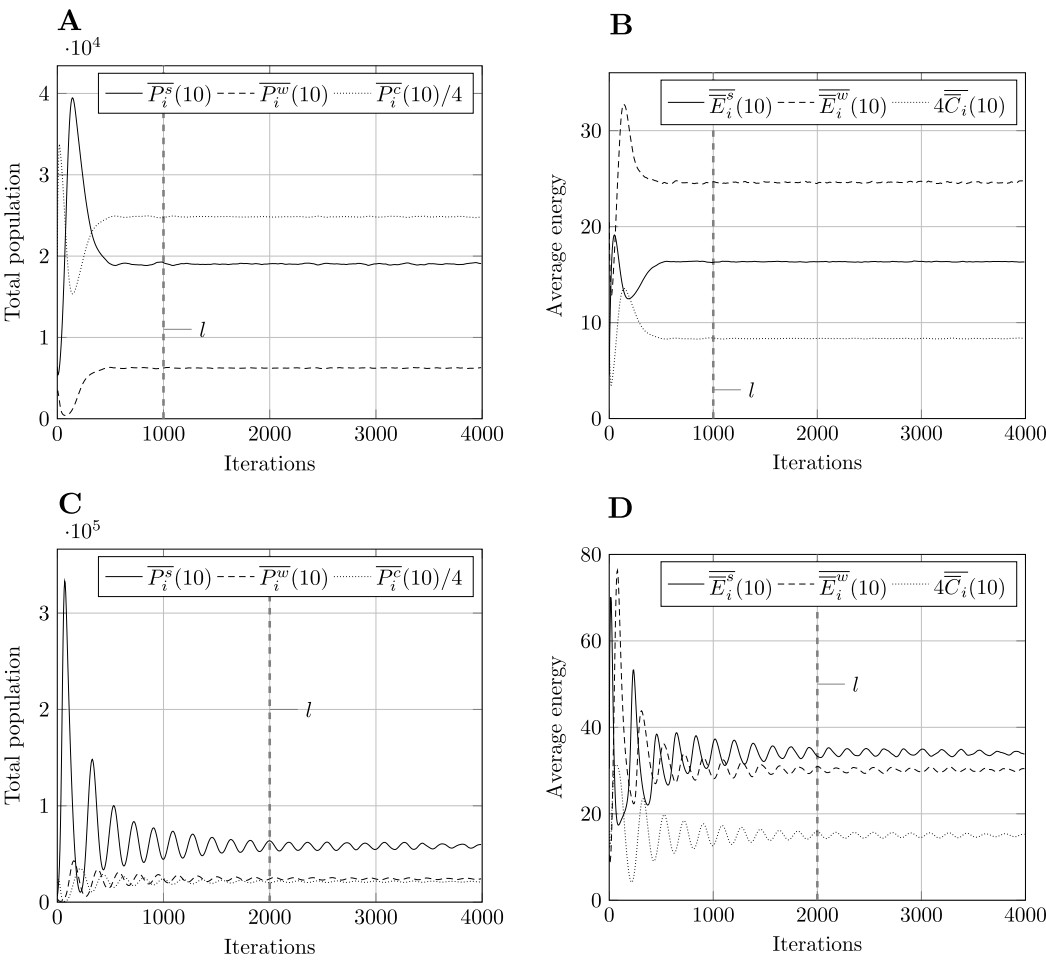

**Figure 3  Moving average of outputs for model size 400 with $w = 10$.** Other model sizes produce similar results, apart from a vertical scaling factor. The dashed vertical line corresponds to iteration $l$ after which the output is considered to be in steady-state. (A) Population moving average, param. set 1. (B) Energy moving average, param. set 1. (C) Population moving average, param. set 2. (D) Energy moving average, param. set 2.

'Methodology', we select the steady-state truncation point to be $l = 1{,}000$ for parameter set 1, and $l = 2{,}000$ for parameter set 2. These are round values which appear to occur after the transient stage. Other model sizes produce similar results, apart from a vertical scaling factor, which means that these values of $l$ are also applicable in those cases.

## Analyzing the distributions of focal measures

The six statistic summaries for each FM, namely mean, sample variance, $p$-value of the SW test, skewness, histogram and Q–Q plot, are shown in Tables S2.1–S2.10 for all model size and parameter set combinations. The number of bins in the histograms is set to the minimum between 10 (an appropriate value for a sample size of 30) and the number of unique values in the data set.

Much of the information provided in Tables S2.1–S2.10, namely the $p$-value of the SW test, the skewness, and the Q–Q plots, is geared towards continuous distributions.

**Table 8  Histograms for the several size@set combinations of the arg max$P_i^s$ FM.**

| Set | Size | | | | |
|---|---|---|---|---|---|
|  | 100 | 200 | 400 | 800 | 1,600 |
| 1 |  |  |  |  |  |
| 2 |  |  |  |  |  |

However, FMs taken from *arg max* and *arg min* operators only yield integer (discrete) values, which correspond to specific iterations. The same is true for *max* and *min* of population outputs, namely $P_i^s$, $P_i^w$, and $P_i^c$. This can be problematic for statistic summaries taken from integer-valued FMs with a small number of unique values. For example, the SW test will not be very informative in such cases, and cannot even be performed if all observations yield the same value (e.g., *arg max* of $P_i^c$ for 800@1, Table S2.4). Nonetheless, distributional properties of a FM can dramatically change for different model size and parameter set combinations. For example, for parameter set 2, observations of the *arg max* of $P_i^c$ span many different values for model size 200 (Table S2.7), while for size 1,600 (Table S2.10) they are limited to only three different values. Summary statistics appropriate for continuous distributions could be used in the former case, but do not provide overly useful information in the latter. In order to maintain a consistent approach, our discussion will continue mainly from a continuous distribution perspective, more specifically by analyzing how closely a given FM follows the normal distribution, though we superficially examine its discrete nature when relevant.

### Distribution of focal measures over the several size@set combinations

In the next paragraphs we describe the distributional behavior of each FM, and when useful, repeat in a compact fashion some of the information provided in Tables S2.1–S2.10.

max$P_i^s$.  The SW *p*-value is consistently above the 5% significance level, skewness is usually low and with an undefined trend, and the Q–Q plots mostly follow the $y = x$ line. Although there are borderline cases, such as 800@1 and 1,600@2, the summary statistics show that the maximum prey population FM generally follows an approximately normal distribution.

arg max$P_i^s$.  This FM follows an approximately normal distribution for smaller sizes of parameter set 1, but as model size grows larger, the discrete nature of the data clearly stands out. This behavior is more pronounced for parameter set 2 (which yields simulations inherently larger than parameter set 1), such that, for 1,600@2, all observations yield the same value (i.e., 70). Table 8 shows, using histograms, how the distribution qualitatively evolves over the several size@set combinations.

min$P_i^s$.  Two very different behaviors are observed for the two parameter sets. In the case of parameter set 1, this FM has a slightly negatively skewed distribution, with some *p*-values below the 0.05 significance threshold, but is otherwise not very far from normality

**Table 9 Q–Q plots for the several size@set combinations of the arg max$P_i^w$ FM.**

| Set | Size | | | | |
|---|---|---|---|---|---|
| | 100 | 200 | 400 | 800 | 1,600 |
| 1 | | | | | |
| 2 | | | | | |

(this is quite visible in some histograms). However, for parameter set 2, the data is more concentrated on a single value, more so for larger sizes. Note that this single value is the initial number of prey, which means that, in most cases, the minimum number of prey never drops below its initial value.

arg min$P_i^s$. This FM follows a similar pattern to the previous one, but more pronounced in terms of discreteness, namely for parameter set 1. For parameter set 2, sizes 100 and 200, the distribution is bimodal, with the minimum prey population occurring at iteration zero (i.e. initial state) or around iteration 200, while for larger sizes, the minimum always occurs at iteration zero.

$\overline{P_i^{ss}}$. The prey population steady-state mean seems to generally follow a normal distribution, the only exception being 400@2, in which some departure from normality is observed, as denoted by a SW $p$-value below 0.05 and a few outliers in the Q–Q plot.

$S^{ss}(P_i^s)$. For most size@set combinations this FM does not present large departures from normality. However, skewness is always positive.

max$P_i^w$. This FM presents distributions which are either considerably skewed or relatively normal. The former tend to occur for smaller model sizes, while the latter for larger sizes, although this trend is not totally clear. The 800@2 sample is a notable case, as it closely follows a normal distribution, with a symmetric histogram, approximately linear Q–Q plot, and a SW $p$-value of 0.987.

arg max$P_i^w$. Interestingly, for parameter set 1, this FM seems to follow a uniform distribution. This is more or less visible in the histograms, but also in the Q–Q plots, because when we plot uniform data points against a theoretical normal distribution in a Q–Q plot we get the "stretched-S" pattern which is visible in this case (Table 9). For parameter set 2, the distribution seems to be more normal, or even binomial as the discreteness of the data starts to stand-out for larger model sizes; the only exception is for size 100, which presents a multimodal distribution.

min$P_i^w$. The minimum predator population seems to follow an approximately normal distribution, albeit with a slight positive skewness, except for 800@1, which has negative skewness.

$\arg\min P_i^w$. This FM displays an approximately normal distribution. However, for larger simulations (i.e. mainly for parameter set 2) the discrete nature of the data becomes more apparent.

$\overline{P_i^w}^{ss}$. The steady-state mean of predator population apparently follows a normal distribution. This is confirmed by all summary statistics, such as the SW $p$-value, which is above 0.05 for all size@set combinations.

$S^{ss}(P_i^w)$. Departure from normality is not large in most cases (200@2 and 800@2 are exceptions, although the former due to a single outlier), but the trend of positive skewness is again observed for this statistic.

$\max P_i^c$. The maximum available cell-bound food seems to have a normal distribution, although 400@2 has a few outliers which affect the result of the SW $p$-value (which, nonetheless, is above 0.05).

$\arg\max P_i^c$. The behavior of this FM is again quite different between parameter sets. For the first parameter set, the discrete nature of the underlying distribution stands out, with no more than three unique values for size 100, down to a single value for larger sizes, always centered around the value 12 (i.e. the maximum available cell-bound food tends to occur at iteration 12). For the second parameter set, distribution is almost normal for sizes above 200, centered around iteration 218, although its discreteness shows for larger sizes, namely for size 1,600, which only presents three distinct values. For size 100, most values fall in iteration 346, although two outliers push the mean up to 369.5.

$\min P_i^c$. This FM displays an apparently normal distribution for all model sizes and parameter sets, with the exception of 800@1, which has a few outliers at both tails of the distribution, bringing down the SW $p$-value barely above the 5% significance level.

$\arg\min P_i^c$. In this case, the trend is similar for both parameter sets, i.e. the distribution seems almost normal, but for larger sizes the underlying discreteness becomes apparent. This is quite clear for parameter set 2, as shown in Table 10, where the SW test $p$-value decreases as the discreteness becomes more visible in the histograms and Q–Q plots.

$\overline{P_i^c}^{ss}$. For this FM there is not a significant departure from normality. The only exception is for 800@1, but only due to a single outlier.

$S^{ss}(P_i^c)$. Like in previous cases, the steady-state sample standard deviation does not stray too far from normality, but consistently shows a positive skewness.

$\max \overline{E_i^s}$. For sizes 100 and 200 of both parameter sets, the maximum of the mean prey energy presents a positively skewed, lognormal-like distribution. For larger sizes, distributions tend to be more normal-like. This trend is clear when analyzing how the $p$-value of the SW test and the skewness vary for the several size@set combinations, as shown in Table 11, namely for sizes 100 and 200, where the former is smaller while the absolute value of the latter is larger.

**Table 10 Three statistical summaries for the several sizes of the arg min$P_i^c$ FM for parameter set 2.** Row 'SW' contains the SW test $p$-values, while the corresponding histograms and Q–Q plots are in rows 'Hist.' and 'Q–Q', respectively.

| Set | Size | | | | |
|---|---|---|---|---|---|
| | 100 | 200 | 400 | 800 | 1,600 |
| SW | 0.437 | 0.071 | 0.062 | 0.011 | <0.001 |
| Hist. | | | | | |
| Q–Q | | | | | |

**Table 11 $p$-values for the SW test (row 'SW') and skewness (row 'Skew.') for the several size@set combinations of the max$\overline{E}_i^s$ FM.**

| Set | Stat. | Size | | | | |
|---|---|---|---|---|---|---|
| | | 100 | 200 | 400 | 800 | 1,600 |
| 1 | SW | 0.159 | 0.012 | 0.625 | 0.672 | 0.555 |
| | Skew. | 0.679 | 0.961 | 0.521 | −0.123 | 0.196 |
| 2 | SW | <0.001 | 0.038 | 0.515 | 0.702 | 0.337 |
| | Skew. | 1.80 | 1.07 | −0.327 | −0.216 | 0.389 |

arg max$\overline{E}_i^s$. For parameter set 1, the distribution is approximately normal for smaller sizes, with the underlying discreteness becoming apparent for larger sizes, centering around iteration 49. For parameter set 2, the data set revolves around a limited set of unique values (centered at iteration 16), following a poisson-like distribution, except for size 100, which displays a bimodal behavior.

min$\overline{E}_i^s$. This FM seems to follow an approximately normal distribution.

arg min$\overline{E}_i^s$. In the case of parameter set 1, this FM has distributions with a single value: zero. This means that the minimum mean prey energy occurs at the initial state of the simulation. From there onwards, mean prey energy is always higher. The situation is notably different for the second parameter set, where minimum mean prey energy can occur at several different iterations centered around iteration 88. Distribution seems to be binomial or Poisson-like.

$\overline{\overline{E}}_i^{ss}$. Although the histograms are not very clear, the Q–Q plots and the $p$-values from the SW test suggest that this FM follows a normal distribution.

$S^{ss}(\overline{E}_i^s)$. This FM does not seem to stray much from normality, except in the case of 1,600@1 and 200@2, which are affected by outliers. The tendency for the steady-state sample standard deviation statistic to show positive skewness is again confirmed with these observations (800@1 being the exception).

$\max \overline{E}_i^w$. The maximum of mean predator energy follows an approximately normal distribution, though for 100@1 there are a few replications which produce unexpected results.

$\arg \max \overline{E}_i^w$. In most cases, this FM approximately follows a normal distribution. There are several exceptions though. For the second parameter set and sizes above 400, the FM starts to display its discrete behavior, following a Poisson-like distribution. Less critically, an outlier "ruins" normality for 100@1.

$\min \overline{E}_i^w$. Apart from a few outliers with some parameter combinations, this FM generally seems to follow a normal distribution.

$\arg \min \overline{E}_i^w$. Perhaps with the exception of 100@1 and 200@1, the iteration where the minimum of mean predator energy occurs seems best described with a discrete, Poisson-like distribution.

$\overline{\overline{E}_i^w}^{ss}$. This FM generally follows a normal distribution. However, 1,600@1 shows a salient second peak (to the right of the histogram, also visible in the Q–Q plot), affecting the resulting SW $p$-value, which is below the 1% significance threshold.

$S^{ss}(\overline{E}_i^w)$. This FM follows a positively skewed unimodal distribution, in the same line as the steady-state sample standard deviation of other outputs. Note the outlier in 200@2, also observed for the $S^{ss}(P_i^w)$ FM, which is to be excepted as both FMs are related to predator dynamics.

$\max \overline{C}_i$. The samples representing the maximum of the mean $C$ state variable are most likely drawn from a normal distribution. Most histograms are fairly symmetric (which is corroborated by the low skewness values), the Q–Q plots are generally linear, and the SW $p$-value never drops below 0.05 significance.

$\arg \max \overline{C}_i$. For smaller model sizes this FM follows a mostly normal distribution, but as with other iteration-based FMs, the underlying discreteness of the distribution starts to show at larger model sizes, especially for the second parameter set.

$\min \overline{C}_i$. For most size@set combinations, the minimum of the mean $C$ state variable seems to be normally distributed. Nonetheless, a number of observations for 400@2 yield unexpected values, making the respective distribution bimodal and distorting its normality (though the respective SW $p$-value does not drop below 0.05).

$\arg \min \overline{C}_i$. As in some previous cases, this FM displays different behavior depending on the parameter set. For the first parameter set, practically all observations have the same value, 10, which means the minimum of the mean $C$ state variable is obtained at iteration 10. Only model sizes 100 and 200 have some observations representing iterations 11 and/or 12. Parameter set 2 yields a different dynamic, with an average iteration of 216 approximately (except for size 100, which has an average iteration of 373.3 due to a few very

**Table 12 Empirical classification (from 0 to 5) of each FM according to how close it follows the normal distribution for the tested size@set combinations.** The last row outlines the overall normality of each statistic.

| $X_i$ | Stat. | | | | | |
|---|---|---|---|---|---|---|
| | $\max_{0 \leq i \leq m} X_i$ | $\arg\max_{0 \leq i \leq m} X_i$ | $\min_{0 \leq i \leq m} X_i$ | $\arg\min_{0 \leq i \leq m} X_i$ | $\overline{X}^{ss}$ | $S^{ss}$ |
| $P_i^s$ | ●●●●● | ●●○○○ | ●●◐○○ | ○○○○○ | ●●●●● | ●●●●◐ |
| $P_i^w$ | ●●●●○ | ●◐○○○ | ●●●●● | ●●●○○ | ●●●●● | ●●●●◐ |
| $P_i^c$ | ●●●●● | ◐○○○○ | ●●●●● | ●●●◐○ | ●●●●● | ●●●●◐ |
| $\overline{E}_i^s$ | ●●●●○ | ●○○○○ | ●●●●● | ◐○○○○ | ●●●●● | ●●●●◐ |
| $\overline{E}_i^w$ | ●●●●● | ●●●○○ | ●●●●● | ◐○○○○ | ●●●●● | ●●●●○ |
| $\overline{C}_i$ | ●●●●● | ●●◐○○ | ●●●●● | ○○○○○ | ●●●●● | ●●●●◐ |
| Stat. wise | ●●●●◐ | ●●○○○ | ●●●●◐ | ●○○○○ | ●●●●● | ●●●●◐ |

distant outliers). While sizes 200 and 400 follow an approximately normal distribution, larger sizes seem more fit to be analyzed using discrete distributions such as Poisson or binomial.

$\overline{\overline{C}}_i^{ss}$. This FM follows an approximately normal distribution. While most size/parameter combinations have a few outliers, only for 800@1 is the existing outlier capable of making the SW test produce a $p$-value below the 5% significance threshold.

$S^{ss}(\overline{C}_i)$. Although passing the SW normality test ($p$-value > 0.05) in most cases, we again note the positive skewness of the steady-state sample standard deviation samples, suggesting that distributions such as Weibull or Lognormal maybe a better fit.

### Statistics-wise distribution trends

Table 12 summarizes the descriptions given in the previous section. It was built by assigning an empirical classification from 0 to 5 to each FM according to how close it follows the normal distribution for the tested size@set combinations. More specifically, individual classifications were determined by analyzing the information provided in Tables S2.1–S2.10, prioritizing the SW test result (i.e. if the $p$-value is above 0.01 and/or 0.05) and distributional discreteness (observable in the Q–Q plots). This classification can be used as a guide to whether parametric or non-parametric statistical methods should be used to further analyze the FMs or to compare FMs of different PPHPC implementations. The last row shows the average classification of individual outputs for a given statistic, outlining its overall normality.

The *max* and *min* statistics yield mostly normal distributions, although care should be taken when the maximum or minimum systematically converge to the same value, e.g., when they occur at iteration zero. Nonetheless, parametric methods seem adequate for FMs drawn from these statistics. The same does not apply to the *arg max* and *arg min* statistics, which show a large variety of distributional behaviors (including normality in some cases). Thus, these statistics are better handled with non-parametric techniques. The steady-state mean typically displays distributions very close to normal, probably due to central-limit-theorem type effects, as described by *Law (2007)* for mean or average-based

FMs. Consequently, parametric methods will most likely be suitable for this statistic. Finally, FMs based on the steady-state sample standard deviation display normal-like behavior, albeit with consistently positive skewness; in fact, they are probably better represented by a Weibull or Lognormal distribution. While parametric methods may be used for this statistic, results should be interpreted cautiously.

## DISCUSSION

In this paper, the PPHPC model is completely specified, and an exhaustive analysis of the respective simulation outputs is performed. Regarding the latter, after determining the mean and variance of the several FMs, we opted to study their distributional properties instead of proceeding with the classical analysis suggested by simulation output analysis literature (i.e., the establishment of CIs.). This approach has a number of practical uses. For example, if we were to estimate CIs for FMs drawn from the steady-state mean, we could use $t$-distribution CIs with some confidence, as these FMs display an approximately normal distribution. If we did the same for FMs drawn from the steady-state sample standard deviation, the *Willink (2005)* CI would be preferable, as it accounts for the skewness displayed by these FMs. Estimating CIs without a good understanding of the underlying distribution can be misleading, especially if the distribution is multimodal. The approach taken here is also useful for comparing different PPHPC implementations. If we were to compare *max* or *min*-based FMs, which seem to follow approximately normal distributions, parametric tests such as the $t$-test would most likely produce valid conclusions. On the other hand, if we compare *arg max* or *arg min*-based FMs, non-parametric tests, such as the Mann-Whitney $U$ test (*Gibbons & Chakraborti, 2011*), would be more adequate, as these FMs do not usually follow a normal distribution.

However, the scope of the PPHPC model is significantly broader. For example, in *Fachada et al. (2015b)*, PPHPC is reimplemented in Java with several user-selectable parallelization strategies. The goal is to clarify which are the best parallelization approaches for SABMs in general. A $n$-sample statistical test is applied to each FM, for all implementations and strategies simultaneously, in order to verify that these do not yield dissimilar results. In *Fachada et al. (2015a)*, PPHPC is used for presenting a novel model-independent comparison technique which directly uses simulation outputs, bypassing the need of selecting model-specific FMs.

The PPHPC model is made available to other researchers via the source code, in addition to the specification presented here. All the data analyzed in this paper is also available as Supplemental Information. PPHPC can be used as a pure computational model without worrying with aspects like visualization and user interfaces, allowing for direct performance comparison of different implementations.

## CONCLUSION

In this paper, we presented PPHPC, a conceptual model which captures important characteristics of SABMs. The model was comprehensively described using the ODD protocol, a NetLogo canonical implementation was reported, and simulation outputs were thoroughly studied from a statistical perspective for two parameter sets and several model sizes. While

many ABMs have been published, proper model description and analysis is lacking in the scientific literature, and thus this paper can be seen as a guideline or methodology to improve model specification and communication in the field. Furthermore, PPHPC aims to be a standard model for research in agent-based modeling and simulation, such as, but not limited to, statistical model comparison techniques, performance comparison of parallel implementations, and testing the influence of different PRNGs on the statistical accuracy of simulation output.

### Funding

This work was supported by the Fundação para a Ciência e a Tecnologia (FCT) projects UID/EEA/50009/2013, UID/MAT/04561/2013 and (P. RD0389) Incentivo/EEI/LA0009/2014, and partially funded with grant SFRH/BD/48310/2008, also from FCT. The author Vitor V. Lopes acknowledges the financial support from the Prometeo project of SENESCYT (Ecuador). The funders had no role in study design, data collection and analysis, decision to publish, or preparation of the manuscript.

### Grant Disclosures

The following grant information was disclosed by the authors:
Fundação para a Ciência e a Tecnologia (FCT): UID/EEA/50009/2013, UID/MAT/04561/2013, Incentivo/EEI/LA0009/2014, SFRH/BD/48310/2008.
SENESCYT: Prometeo project.

### Competing Interests

The authors declare there are no competing interests.

### Author Contributions

- Nuno Fachada conceived and designed the experiments, performed the experiments, analyzed the data, wrote the paper, prepared figures and/or tables, performed the computation work, reviewed drafts of the paper.
- Vitor V. Lopes conceived and designed the experiments, wrote the paper, reviewed drafts of the paper.
- Rui C. Martins analyzed the data, reviewed drafts of the paper.
- Agostinho C. Rosa contributed reagents/materials/analysis tools, reviewed drafts of the paper.

### Data Availability

https://github.com/fakenmc/pphpc/tree/netlogo.

### Supplemental Information

Supplemental information for this article can be found online at http://dx.doi.org/10.7717/peerj-cs.36#supplemental-information.

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
