# Peer review of "Towards a standard model for research in agent-based modeling and simulation"

_PeerJ Computer Science, doi:10.7717/peerj-cs.36_

## Round 0.1 · original submission · Major Revisions

This paper presented a template model for agent-based simulations. The proposed PPHPC conceptual model is really needed in ABMs, especially the last three points that authors mentioned. However, the entire paper doesn't complete show the concept "template" very well. Some additional comments are provided below.

1. When authors talk about the necessity of their work and mentioned the currently used ODD's disadvantages, they didn't point out how they overcome them. In addition, they argue that ODD needs one additional section for statistical analysis. But in this paper, they need the additional section too. It doesn't how they cohesively integrate the section into their model.

2. The paper serves as a template. However, when authors present the template, they incorporate the sample model into the template. Thus, the template is not described independently and in very formal way. They need to reorganized the section to make the template sample-independent.

4. The used sample model is widely used in many ABM materials, including NetLogo Manual provided by the development team. I don't see any reason that the very similar sample needs to be described in the scientific paper except a tutorial. If the sample is really needed, reasonable details and the connections to templates need to be carefully presented.

5. It seems the last section is the main contribution of this paper although most of the proposed measures are from other's work.

To summarize, the PPHPC model is interesting, but the tutorial is difficult to present to avoid duplicate work. Authors need to think about their unique contributions and organize the paper according to it. Currently it seems that several things are grabbed and the paper lacks cohesion.

Reviewer 1 ·

Basic reporting

Most of the part is very good, suggest the "A NetLogo implementation" should be a new section.

Experimental design

No comments

Validity of the findings

The Table 12 should give more information to explain, like how to get the Statistics-wise ( for example the value of max Xi is 4.5)

Additional comments

Overall, this is a good paper can give the people a tutorial fashion who want to start his agent-based modeling research. The whole paper used most of paragraphs to show the model's description and analysis. The analysis is the key point of this paper( the author also believe "this paper proposes a methodology on how to achieve these goals), so, the author should add one paragraph in end of the "Background" section, compare to the literature review paper, summarize the shortage of the current ABMs and point out why this work is important and significant.

·

Basic reporting

No Comments

Experimental design

No Comments

Validity of the findings

No Comments

Additional comments

The article, “A template model for agent-based simulations”, proposes an agent-based model named “Predator-Prey for High-Performance Computing”. Experiments using NetLogo implementation are carried out, and results are statistically analyzed. The authors propose to set this model and the corresponding result analyzing process as a paradigm for agent-based modeling research. I think the idea is interesting, and suggest publishing this article.

Reviewer 3 ·

Basic reporting

The reason to select predator-prey model as the template for formal description and output analysis is not discussed, although many other alternatives (StupidModel/Sugarscape/Heatbugs/Boids) are mentioned. A discussion on choosing predator-prey model needs to be conducted.

Experimental design

Line 36 “While no formal standard for ABM description exists, the ODD protocol (Overview, Design concepts, Details) is currently one of the most widely used templates…” It’s a not a convincing explanation for readers to understand why you would use ODD protocol for model conceptual description, as there does exist popular formalisms to describe conceptual models in classic M&S area, such as DEVS.

Validity of the findings

No Comments

Additional comments

Minor problems:
1 Line 20 Lack of reference
2 Line 313 Double “can be”
3 Algorithm 2 can be discarded.

Reviewer 4 ·

Basic reporting

No Comments

Experimental design

No Comments

Validity of the findings

No Comments

Additional comments

The paper presents a conceptual model which captures important characteristics of spatial agent-based models. The model is elegantly described and evaluated its correctness by a small example implemented in Netlogo. A comprehensive simulation output analysis theory is reviewed and illustrated by the simulation data of the example.
There are two small problems which should be revised based on my understanding:
1. In table 1 (page 4), the reproduction probability should be real numbers in [0,1], not integers. Also the variables are not very clear, for example, based on the row of Grid cells, X and Y should be the size of the whole cell space environment; but based on the row of Agents, X and Y are interpreted as the position of an agent. Please be more consistent and clear.
2. In the initialization process of page 5, the procedure describes only one agent’s initialization, please give a complete procedure for multiple agents, though the procedure is very simple.

---

## Round 0.2 · accepted · Accept

The authors changed the title and revised the paper accordingly to keep the consistency. I recommend the paper be accepted.

·

Basic reporting

No comments

Experimental design

No comments

Validity of the findings

No comments

Additional comments

The paper is improved according to reviewers' comments from the first round. The motivation and the proposed "Predator-Prey for High-Performance Computing" model become clearer. I again suggest publishing it.

Reviewer 3 ·

Basic reporting

No Comments

Experimental design

No Comments

Validity of the findings

No Comments

Additional comments

No Comments

Reviewer 4 ·

Basic reporting

No comments.

Experimental design

No comments.

Validity of the findings

No comments.

Additional comments

Based on the first round review, the author has revised the paper accordingly, and it is much more clear and consistent.